# Molecular Genetic Analysis of Russian Patients with Coagulation Factor FVII Deficiency

**DOI:** 10.3390/genes14091767

**Published:** 2023-09-06

**Authors:** Olesya Pshenichnikova, Daria Selivanova, Ekaterina Shchemeleva, Tatiana Abramova, Nadezhda Zozulya, Vadim Surin

**Affiliations:** 1Laboratory of Genetic Engineering of National Medical Research Center for Hematology, Novy Zykovski lane 4a, 125167 Moscow, Russia; 2Coagulopathies Department of National Medical Research Center for Hematology, Novy Zykovski lane 4a, 125167 Moscow, Russia

**Keywords:** FVII deficiency, *F7* gene, rare bleeding disorder, molecular genetic analysis

## Abstract

Coagulation factor VII (proconvertin) is one of the proteins starting the blood coagulation cascade. Plasma FVII concentration is regulated by different factors. A low level of FVII could also be a result of FVII deficiency (MIM# 227500), the rare autosomal recessive inherited disease caused by pathogenic variants in the *F7* gene. The aim of this study was to describe a mutation spectrum of the *F7* gene and genotype–phenotype relationship in patients with FVII deficiency in Russia for the first time. We studied the primary structure of the *F7* gene of 54 unrelated patients with FVII deficiency by direct Sanger sequencing. Pathogenic variants in the *F7* gene were detected in 37 (68.5%) of them. We identified 24 different mutations located mostly in the serine protease domain. Five pathogenic variants had never been reported before. A major mutation in the Russian population was c.1391delC (p. Pro464Hisfs*32), linked with rs36209567 and rs6046 functional polymorphisms, that is widely distributed in East Europe. As in other countries, the *F7* genotypes poorly correlated with the severity of clinical manifestations but were quite well associated with FVII levels. Minor alleles of functional polymorphisms rs510335, rs5742910, rs561241, rs36209567, and rs6046 could also participate in the *F7* genotype and influence FVII levels.

## 1. Introduction

The blood coagulation process starts from the formation of the complex between the tissue factor (TF) and the activated coagulation factor FVII (FVIIa) when the vascular endothelium is damaged. A complete lack of any of these proteins is incompatible with life and leads to death in early childhood because of severe hemorrhages. However, even small amounts of them are enough to initiate a number of reactions on membrane surfaces during complex TF-FVIIa formation [1].

FVII is a zymogen of vitamin-K-dependent serine protease FVIIa, which is synthesized mostly in the liver and circulates in plasma. Plasma FVII concentration is regulated by different factors such as age, body mass index, dietary fat intake, plasma lipids (especially triglycerides), diabetes, and *F7* polymorphisms [2,3].

The *F7* gene (MIM# 613878) coding the FVII protein is located on chromosome 13 (13q34), consists of nine exons, and has a length of about 12.8 kbp. There are three transcripts of the gene resulting from alternative splicing. Two transcripts (NM_019616.4, NM_000131.4) encode identical mature proteins circulating in plasma, and in the third transcript (NM_001267554.1), terminal domains are absent, and its physiological role is unknown [1]. mRNA encodes 466 amino acids; the first twenty encode the signal peptide, the next forty—propeptide, providing biosynthesis/secretion of FVII, and these 60 amino acids are removed in the process of intracellular proteolysis. The mature FVII protein (50 kDa) consists of 406 amino acids. The 1–406 numeration is more widely used in protein description, while the 1–466 numeration is used mainly in molecular genetics; therefore, we will use the second one. The mature protein contains a light and a heavy chain. The light chain consists of a calcium-binding, vitamin-K-dependent Gla domain (61–142 amino acid residues) and two EGF-like domains (106–142 and 143–211 amino acid residues) interacting with membranes and other proteins of the coagulation cascade. The heavy chain contains a serine protease domain (212–466 amino acid residues) [4]. FVII activation occurs during proteolytic cleavage between Arg212-Ile213 amino acid residues [1,4].

Pathogenic variants in the *F7* gene lead to the rare autosomal recessive inherited disease known as FVII deficiency (MIM# 227500) with a worldwide prevalence of 1:300,000–1:500,000. Nowadays, 271 unique variants in 1058 individuals are described in the international database EAHAD Factor VII Gene (*F7*) Variant Database (https://f7-db.eahad.org/ (accessed on 23 August 2023)). Patients with FVII deficiency demonstrate a high phenotypic variability, from asymptomatic to life-threatening gastrointestinal bleedings and central nervous system hemorrhages. Most patients have mucosal bleedings (epistaxis, gum bleeding, easy bruising), and women often present with menorrhagia. Despite active attempts to study the molecular basis of FVII deficiency, there has been no significant correlation between the *F7* genotype, FVII factor activity (FVII:C), and a clinical phenotype found [5,6,7,8,9]. The genetic impact of *F7* polymorphisms on FVII plasma levels is estimated at 57–63% [2]. Among the most studied single nucleotide polymorphisms (SNP), minor alleles of rs510335 (−401G/T), rs5742910 (−3230 bp/10 bp) in the promoter gene region, and rs6046 (Arg353Gln) in exon 9 are consistently associated with lower plasma FVII levels [3,10,11]. Nevertheless, because of the disease’s rarity, it is extremely important to collect information on patients with FVII deficiency from different populations, as it will allow for obtaining enough data for more detailed analysis and pattern findings.

In Russia, molecular genetic analysis of the *F7* gene is conducted predominantly in our center. Therefore, the aim of this study was to describe a mutation spectrum of the gene in our country and the genotype–phenotype relationship in patients with FVII deficiency.

## 2. Materials and Methods

In this study, we included 54 unrelated patients (12 males, 42 females) with declined levels of FVII and a presumptive diagnosis of FVII deficiency (less than 70% FVII:C) with different types of severity. In our country, we have issues with diagnostics of rare diseases in regions. The majority of studied patients were from Moscow and the Moscow region (N = 35, 65%), while other 14 patients (26%) were from European part of Russia, and only 5 patients (9%) represented other regions. The collaborating physicians at our center who consulted and selected these patients were experienced in the diagnosis and treatment of hemophilia A, B, von Willebrand disease, and rare bleeding disorders. The study was carried out according to the Principles of the Declaration of Helsinki; informed consent was obtained from all participants.

Genomic DNA was isolated from EDTA-treated whole blood samples using phenol–chloroform extraction and ethanol precipitation [12]. Genomic DNA was dissolved in TE buffer (pH 8.0) and frozen at −20 °C until genotyping.

To conduct molecular genetic analysis, we amplified all functionally important regions of *F7* gene (466 bp of promoter, exons 1–9, and exon-intron junctions) as 8 fragments with length of 261–775 bp with primers designed in our laboratory according to the reference genomic *F7* sequence NG_009262 (Table 1, Figure 1).

PCR reactions were carried out on a Tercik programmable thermocycler (DNK-Technology, Moscow, Russia) with PCR Master Mix (Thermo Fisher Scientific, Waltham, MA, USA) using 10 pmol each oligonucleotide primer (Syntol, Moscow, Russia) and 50–100 ng template DNA in 25 μL of reaction mixture. Amplification conditions for these primer sets were 35 cycles of 94 °C for 1 min, 62 °C for 1 min, 72 °C for 3 min. We analyzed the obtained PCR fragments with electrophoresis in 6% polyacrylamide gel (PAAG) followed by staining with ethidium bromide (0.03 µg/mL) for 1 min and visualization in UV light. Amplified DNA fragments were purified using Wizard PCR Preps DNA Purification System (Promega Corporation, Madison, WI, USA) and subjected to direct Sanger sequencing using ABI PRISM BigDyeTM Terminator v.3.1 Cycle Sequencing Kit (Thermo Fisher Scientific, Waltham, MA, USA) on ABI PRISM 3100Avant genetic analyzer sequencer (Applied Biosystems, Foster City, CA, USA) at Genome CCU (Institute of Molecular Biology, Russian Academy of Sciences, Moscow, Russia).

The cDNA numbering system used is compliant with the Human Genome Variation Society recommendations ver. 15.11 (http://varnomen.hgvs.org (accessed on 23 August 2023)). The amino acid numbering is based upon the start methionine, codon +1; the reference sequence used was NG_009262 for genomic positioning and NM_000131.4 for cDNA numbering. As reference databases for pathogenic variants, we used Factor VII Variant Database (https://f7-db.eahad.org/ (accessed on 23 August 2023)), Human Gene Mutation Database (www.hgmd.cf.ac.uk (accessed on 23 August 2023)), and ClinVar (https://www.ncbi.nlm.nih.gov/clinvar/ (accessed on 23 August 2023)).

Missense variants unreported in databases were examined using deleteriousness prediction scoring programs: PolyPhen-2 v2.2.2 [13], MutationTaster2 (build NCBI 37/Ensembl 69 [14]), FATHMM (v2.3) [15]. Variant effects were then classified using the ACMG/AMP Variant Curation Guidelines [16].

All statistical analyses were performed using R Statistical Software (v.4.3.0 [17]). Differences in SNP frequencies between our and global populations were estimated with chi-square test with *p*-value < 0.05 considered statistically significant. Hardy–Weinberg equilibrium (HWE) was assessed with chi-square test. Pairwise linkage disequilibrium (LD) between SNPs was calculated from the genotype data and measured as D’. HWE and LD were calculated using the package “genetics” [18]).

## 3. Results

Molecular genetic analysis allowed us to identify pathogenic variants in the *F7* gene in 37 out of 54 patients with a presumptive diagnosis of FVII deficiency (68.5%). We did not find severe disruption of the *F7* gene despite a decreased level of FVII activity levels in 17 patients. However, in 11 (64.5%) of them, we could not exclude the influence of functional polymorphisms on both alleles. A summary of found *F7* gene defects, functional polymorphisms, and clinical features of patients are presented in Appendix A. Among all studied patients with a decreased level of FVII:C, 25 patients (46.3%) were homozygous or compound heterozygous for *F7* mutations, 12 (22.2%) were heterozygous, whereas in 17 (31.5%), no *F7* pathogenic variants were detected.

### 3.1. F7 Mutation Spectrum

Among 37 patients with identified *F7* gene defects, we found 24 different pathogenic variants, five of which had never been reported before in the international databases (Table 2). The majority of gene defects were single nucleotide variations (21 variants, 87.5%); the other three variants were frameshifts. The most common defect type was missense mutation—there were 16 of them (66.7%), and two of them also had an effect on splicing; there were also three regulatory mutations, one frameshift, two inframe microdeletions, and two splicing mutations (besides the abovementioned two missense variants) (Table 2). Most mutations were found in exon 9, but it could be explained by the greatest size of this exon compared with others. Overall, all found defects were distributed uniformly along the gene (Figure 2). Nevertheless, it is characterized by the predominance of missense mutations located mainly in the serine protease domain.

A major mutation in our population was c.1391delC (p.Pro464Hisfs*32); it was found as 29 alleles in 22 patients (59.5% out of all with pathogenic variants), and six times it was in a homozygous state. The regulatory variant c. −55C>T occurred four times in four people in a heterozygous state. The missense mutation c.64G>A (p.Gly22Ser) appeared in two patients—once, it was homozygous, and in another patient, it was heterozygous. Five gene defects (c.291+1G>A, c.691_693del (p.Leu231del), c.817_831del (p.Leu273_Asp277del), c.911C>T (p.Ala304Val), c.1285G>A (p.Ala429Thr)) occurred twice in two people each. The other 16 pathogenic variants were found only once (Table 2).

### 3.2. Novel Mutations

Among previously undescribed variants, one was a splicing mutation disturbing a canonical splice site (c.347−1G>A), and per the ACMG/AMP Variant Curation Guidelines, it was evaluated as likely pathogenic (PM2, PVS1). The other four were missense variants evaluated as variants of unknown significance by the ACMG criteria. Two of them—c.65G>A (p.Gly22Asp) and c.713G>C (p.Cys238Ser)—affected positions with known pathogenic variants (PM2, PM5, PP3); the other two, c.728 T>C (p.Ile243Thr) and c.1175A>G (p.Tyr392Cys), were completely new (PM2, PP3). Pathogenicity prediction algorithms estimated all previously undescribed missense variants as probably pathogenic.

### 3.3. Functional Polymorphisms

Except for obviously pathogenic variants, we also detected known functional polymorphisms: rs510335 (c. −401G>T), rs5742910 (c. −323ins10), rs561241 (c. −122T>C), rs36209567 (c.1061 C>T (p.Ala354Val)), and rs6046 (c.1238 G>A (p.Arg413Gln)). Although in Russian and global populations, their minor allele frequencies (MAF) are quite similar, the allele frequencies in our sample differed statistically significantly from them (Table 3). All polymorphisms were in Hardy–Weinberg equilibrium (*p* > 0.05) and in linkage disequilibrium (Figure 3), especially rs36209567 and rs6046, which were highly correlated with the pathogenic variant c.1391delC (p.Pro464Hisfs*32). We also noted that rs36209567, rs6046, and c.1391delC rather often were met with minor alleles of neutral polymorphisms rs6039 (c.64+9G>A), rs2774033 (c.291+71G>A), and rs6042 (c.525C>T (p.His175His)).

Polymorphisms rs510335 (c. −401G>T), rs5742910 (c. −323ins10), and rs561241 (c. −122T>C) had a complete allelic association and form a single haplotype (Figure 3).

### 3.4. Genotype and Phenotype Interaction

We estimated a clinical phenotype based on FVII level [29] and clinical manifestations [30].

Clinical phenotype was poorly correlated with the FVII level (Table 4) as well as with the *F7* genotype (Table 5). In our sample, 50% of probands (27 patients) had an FVII:C ≤ 10%, 40.7% (22 patients) had an FVII:C > 20%, and only 9.3% (5 patients) had an intermediate level of FVII activity. As for clinical phenotype, probands with mild severity prevailed (42.6%, 23 patients), with such symptoms as epistaxis, gum bleeding, and easy bruising, although almost one-fifth of the sample was asymptomatic (Table 4, Appendix A).

Proband’s genotype quite exactly reflected FVII level. All people with the homozygous pathogenic variant, almost all heterozygous compounds, and people with a combination of the heterozygous pathogenic variant and homozygosity by functional polymorphisms had a low FVII level. Probands with one heterozygous pathogenic variant but without homozygosity by functional polymorphisms or without a found pathogenic variant had a higher FVII level, mostly >20% (Table 5). The only exception was one proband with no mutation or functional polymorphisms found, almost an absence of symptoms, but a low factor activity level (FVII:C = 3.3%). In this patient, there could be either deep intronic mutations or FVII deficiency, in this case resulting from other mechanisms.

In our sample, females prevailed (42 vs. 12 males). Nevertheless, both sexes had quite similar distributions in FVII levels, the *F7* genotype, and clinical symptoms.

## 4. Discussion

### 4.1. Mutation Spectrum of F7 Gene

The spectrum of pathogenic defects in the *F7* gene in Russian patients with reduced FVII levels is similar to the spectra in other countries. It is characterized by the prevalence of missense mutations located predominantly in the serine protease domain, mainly because of the length of exon 9 but also because of its functional significance [5,6,7,9,31,32]. A major mutation in our population was c.1391delC (p.Pro464Hisfs*32), which is supposed to have its origin in Central Europe [6]. We found only one allele that was not linked with the rs36209567 and rs6046 functional polymorphisms. This combination of c.1391delC (p.Pro464Hisfs*32) with rs36209567 was previously described as common in East Europe (particularly in Slovakia), and it was also found in other European countries but with significantly less frequency [6]. Our sample, however, was biased because of a lack of diagnoses of rare diseases in regions, so we described mostly the mutation spectrum of the European part of the country. An extension of the sample to the Asian part of Russia could increase the variability of pathogenic variants or reveal mutations specific to certain regions.

### 4.2. Gene Variant c.1061 C>T (p.Ala354Val)

Variant rs36209567 (c.1061 C>T (p.Ala354Val)) should be discussed separately. In the literature, there are different opinions about its influence on the FVII protein. In most studies, it is referred to as a mutation [6,7,28,31,33]. Nevertheless, we suppose that it seems to be a functional polymorphism because its minor allele in our sample occurred with a frequency of 0.3889, which is too high for a pathogenic variant and, in most cases (in 31 probands out of 54), it was found in combination with two obviously pathogenic defects. Moreover, in patients homozygous by rs36209567 and without found pathogenic variants, we usually saw FVII:C > 20% (Table 5), indicating only a small reduction in FVII activity. However, combined with heterozygosity by other pathogenic variants, it led to a significant reduction in FVII levels (Appendix A). It is according to the functional study of rs36209567 by Fromovich-Amit and colleagues, which demonstrated that this variant alone or in combination with rs6046 (c.1238 G>A (p.Arg413Gln)) led to the reduction of FVII to 41% and 55%, respectively [31].

### 4.3. Functional Polymorphisms

The influence of other functional polymorphisms—rs510335 (c. −401G>T), rs5742910 (c. −323ins10), rs561241 (c. −122T>C), and rs6046 (c.1238 G>A (p.Arg413Gln))—was also proved by functional studies [10,34,35]. The first three promoter variants have a complete allelic association and form a single haplotype, which reduces a basal level of FVII transcription [34,35]. We also showed that in a homozygous state without other *F7* defects, they led to a reduction in FVII levels compared with the heterozygous influence of pathogenic variants (Table 5, Appendix A).

### 4.4. Genotype and Phenotype Interaction

As in other studies [5,6,7,9,30,31,32], we showed a lack of relationships between the severity of clinical manifestations and genotypes (Table 4 and Table 5). Peyvandi and colleagues conducted a large study of the association between coagulant factor activity and clinical bleeding severity of rare bleeding disorders, which revealed a lack of it for FVII deficiency. They proposed to use the laboratory phenotype for this disease expressed by FVII levels [29]. Our results confirm that FVII levels correlate much better with the *F7* genotype than clinical manifestations. Serious damage to both alleles resulting from a homozygous pathogenic defect, heterozygous compound, or heterozygous pathogenic defect combined with homozygosity by functional polymorphisms leads to a significant decrease in plasma FVII (FVII:C ≤ 10%). One allele damaged by a heterozygous pathogenic defect, or both alleles affected by homozygous functional polymorphisms, lead to much more mild consequences. However, even small amounts of FVII are sufficient to initiate the blood coagulation process [1], so most patients with a reduced level of FVII are asymptomatic or have only mild symptoms, such as epistaxis, gum bleeding, or easy bruising.

### 4.5. Patients without Found Pathogenic Variants

We did not find any pathogenic variants in 17 out of 54 patients with FVII deficiency (31.5%). It is a usual situation in FVII deficiency studies [7,9,36]. In 11 of them (64.7%), lower FVII levels could be explained by homozygous functional polymorphisms that partly reduced FVII. In one of the patients without pathogenic variants or homozygous functional polymorphisms, the diagnosis later was changed to the mild autoimmune acquired combined blood clotting factors FV, FVII, and FXII deficiency. The other five patients without pathogenic variants or homozygous functional polymorphisms needed to be studied further. Especially interesting among them is one proband with a low factor activity level (FVII:C = 3.3%) and almost asymptomatic clinical manifestation. In these patients, there could be deep intronic mutations (although one of a deep intronic hotspot, c.571 +78G>A, we checked in our sample but did not find it in anyone) [36] or large deletions that need the use of the MLPA method and are not accounted for in this study. There could also be other factors leading to FVII deficiency unrelated to the *F7* gene.

## 5. Conclusions

We report herein the results of a molecular genetic analysis in Russian patients with coagulation factor FVII deficiency.

Overall, the mutation spectrum of the *F7* gene reflects tendencies revealed in other populations. A major mutation in our population was c.1391delC (p.Pro464Hisfs*32), linked with rs36209567 and rs6046 functional polymorphisms. As in other countries, *F7* genotypes were poorly correlated with the severity of clinical manifestations but were quite well associated with FVII levels.

Nevertheless, these data are the first attempt at an *F7* genetic analysis in Russia, and they contribute to local and global knowledge of molecular mechanisms of congenital FVII deficiency, which could improve diagnostics of this disorder in our country. Undoubtedly, this research needs to be continued in order to increase coverage of patients with FVII deficiency, to include more patients from the eastern part of the country, and to study patients without found causal variants in more detail.

## Figures and Tables

**Figure 1 genes-14-01767-f001:**
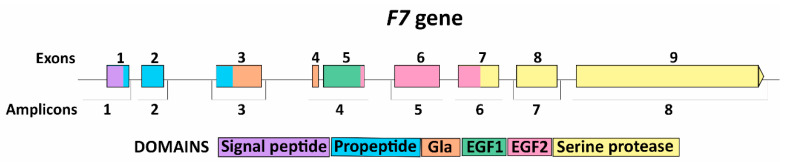
Scheme of amplification system for *F7* gene screening.

**Figure 2 genes-14-01767-f002:**
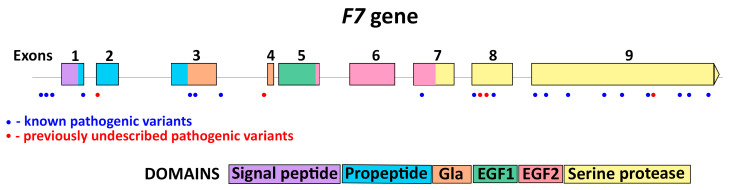
Distribution of different pathogenic variants found among 37 patients from Russia.

**Figure 3 genes-14-01767-f003:**
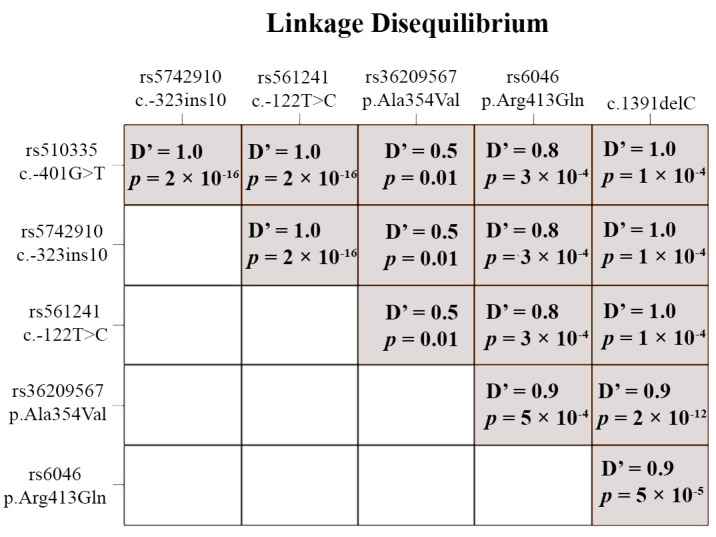
Linkage disequilibrium for functional polymorphisms and c.1391delC (p.Pro464Hisfs*32).

**Table 1 genes-14-01767-t001:** Primers used in the study.

Amplicon	Primer	Nucleotide Sequence	Target	Fragment Length, bp
1	F7F1x	cct aag aaa cca gcc tcc ctt	Promoter; exon 1	686
	F7R1	agc cgc cag aaa acc ctc ct		
2	F7F2	cta gct cac agc atg gcc tt	exon 2	261
	F7R2	cgg tca ctt cct ctc gag ca		
3	F7F3	agc gcc gct ccc ctc ctc ca	exon 3 ^1^	380
	F7R3	gcc gca gcc aaa gag acg ca		
4	F7F45	gtc caa gtc ccc caa ccc ca	exon 4; exon 5	439
	F7R45	cac cta gac caa ttt cca act		
5	F7F6	aga aca cca ctg ctg acc ca	exon 6	384
	F7R6x	tgc cca gat ccc acc tca ca		
6	F7F7	tcc cat agc ctc ggc ctc aa	exon 7	315
	F7R7	cct gcc cat ttt ccc ttc ca		
7	F7F8x	tgc aat gcc aga ggt tcc tt	exon 8 ^1^	477
	F7R8x	gca cga agc cca gag cca ca		
8	F7DF9 F7R9	tgg cca cag ccc atc ccc at	exon 9	775
		cct cct cta ccc cat taa ct		

^1^ Amplification of these fragments was conducted with addition of betaine (5 M, 20% from PCR mix volume).

**Table 2 genes-14-01767-t002:** Pathogenic variants revealed in the study.

#	Pathogenic Variant	Exon/Intron	Domain	Defect Type	Allele Count	Reference
1	c. −55C>T	Promoter	Promoter	Regulatory	4	[19]
2	c. −32A>C	Promoter	Promoter	Regulatory	1	[20]
3	c. −30A>C	Promoter	Promoter	Regulatory	1	[5]
4	c.64G>A (p.Gly22Ser)	Exon 1	Propeptide	Missense/ splicing	3	[21]
5	c.65G>A (p.Gly22Asp)	Exon 2	Propeptide	Missense/ splicing	1	New
6	c.218G>A (p.Leu73Gln)	Exon 3	Gla	Missense	1	[5]
7	c.226G>A (p.Glu76Lys)	Exon 3	Gla	Missense	1	[22]
8	c.291+1G>A	Intron 3	Gla	Splicing	2	[23]
9	c.347−1G>A	Intron 3	Gla	Splicing	1	New
10	c.580C>A (p.Pro194Thr)	Exon 7	EGF2	Missense	1	[24]
11	c.691_693del (p.Leu231del)	Exon 8	Serine protease	Inframe (microdeletion)	2	[6]
12	c.713G>C (p.Cys238Ser)	Exon 8	Serine protease	Missense	1	New
13	c.728 T>C (p.Ile243Thr)	Exon 8	Serine protease	Missense	1	New
14	c.760G>C (p.Cys254Arg)	Exon 8	Serine protease	Missense	1	[6]
15	c.817_831del (p.Leu273_Asp277del)	Exon 9	Serine protease	Inframe (microdeletion)	2	[21]
16	c.847C>T (p.Arg283Trp)	Exon 9	Serine protease	Missense	1	[25]
17	c.911C>T (p.Ala304Val)	Exon 9	Serine protease	Missense	2	[26]
18	c.1072A>G (p.Met358Val)	Exon 9	Serine protease	Missense	1	[25]
19	c.1109G>T (p.Cys370Phe)	Exon 9	Serine protease	Missense	1	[27]
20	c.1171G>A (p.Gly391Ser)	Exon 9	Serine protease	Missense	1	[27]
21	c.1175A>G (p.Tyr392Cys)	Exon 9	Serine protease	Missense	1	New
22	c.1285G>A (p.Ala429Thr)	Exon 9	Serine protease	Missense	2	[22]
23	c.1310A>T (p.Tyr437Phe)	Exon 9	Serine protease	Missense	1	[6]
24	c.1391delC (p.Pro464Hisfs*32)	Exon 9	Serine protease	Frameshift (microdeletion)	29	[28]

**Table 3 genes-14-01767-t003:** Functional polymorphisms revealed in the study, their minor allele frequencies (MAF), and results of chi-square test between global MAF and MAF in our sample.

rs ID	Functional Polymorphism	MAF (Global)	MAF (Russia, RUSeq Data)	MAF (Our Sample)	chi-Square	*p*-Value
rs510335	c. −401G>T	0.1622	No data	0.2593	7.44	0.006
rs5742910	c. −323ins10	0.0523	No data	0.2593	92.9	<0.00001
rs561241	c. −122T>C	0.1128	0.1239	0.2593	22.9	<0.00001
rs36209567	c.1061 C>T (p.Ala354Val)	0.0006	0.0005	0.3889	23472.6	<0.00001
rs6046	c.1238 G>A (p.Arg413Gln)	0.1265	0.1294	0.6111	229.1	<0.00001

**Table 4 genes-14-01767-t004:** Relationship between clinical bleeding severity score based on Peyvandi [29] and Mariani [30].

Clinical Severity Peyvandi (Vertically)/ Mariani (Horizontally)	Severe N (%)	Moderate N (%)	Mild N (%)	Asymptomatic N (%)	No Data N (%)	Total
Severe (FVII:C ≤ 10%)	3 (11.1%)	9 (33.4%)	10 (37%)	4 (14.8%)	1 (3.7%)	27 (50%)
Moderate (10% < FVII:C ≤ 20%)	0	2 (40%)	2 (40%)	1 (20%)	0	5 (9.3%)
Mild (FVII:C > 20%)	0	4 (18.2%)	11 (50%)	5 (22.7%)	2 (9.1%)	22 (40.7%)
Total	3 (5.6%)	15 (27.7%)	23 (42.6%)	10 (18.5%)	3 (5.6%)	

**Table 5 genes-14-01767-t005:** Relationship between proband’s clinical bleeding severity score and genotype.

Pathogenic Variant Status	Total	Clinical Severity Peyvandi [29]			Clinical Severity Mariani [30]				
		Severe (FVII:C ≤ 10%) N (%)	Moderate (10% < FVII:C ≤ 20%) N (%)	Mild (FVII:C > 20%) N (%)	Severe N (%)	Moderate N (%)	Mild N (%)	Asymptomatic N (%)	No Data N (%)
homozygous	8	8 (100%)	0	0	1 (12.5%)	4 (50%)	2 (25%)	0	1 (12.5%)
heterozygous compound	17	15 (88.2%)	2 (11.8%)	0	2 (11.8%)	3 (17.7%)	8 (47%)	4 (23.5%)	0
heterozygous	9	0	1 (11.1%)	8 (88.9%)	0	3 (33.3%)	3 (33.3%)	3 (33.4%)	0
heterozygous + homozygous by functional polymorphisms	3	3 (100%)	0	0	0	2 (66.7%)	0	1 (33.3%)	0
no mutation found + homozygous by functional polymorphisms	12	0	2 (16.7%)	10 (83.3%)	0	2 (16.7%)	7 (58.3%)	1 (8.3%)	2 (16.7%)
no mutation found	5	1 (20%)	0	4 (80%)	0	1 (20%)	3 (60%)	1 (20%)	0
Total	54	27	5	22	3	15	23	10	3

## Data Availability

Data are available on request from the authors.

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
