# Peer review of "Molecular Genetic Analysis of Russian Patients with Coagulation Factor FVII Deficiency"

_genes, 2023, doi:10.3390/genes14091767_

Round 1
Reviewer 1 Report
The manuscript is well organized. It represent a regional database and the results are similar to those globally reported. Definitely, the manuscript must be reported the results disseminated.
Minor aspects
Explain the PCR protocol, we have only one primer for exon 9, and it is wrritted that 8 amplicons were sequenced.
After the Table with the primers include one Figure that display with narrows the amplicons location. You can use the existed figure (Distribution of different pathogenic variants found among 37 patients from Russia.) to make another one. By these will be simple for readers to see the targeted locations.
Author Response
Thank you very much for taking the time to review our manuscript. Please find the detailed responses below and the corresponding corrections in track changes in the resubmitted files.
Thank you for your attention, we made some typos in Table 1, so we corrected them, made more clear division of primers to different amplicons and added a figure you asked for (please see Figure 1, Table 1).
Reviewer 2 Report
This is the first study presented with a small Russian patient cohort to determine the mutation spectrum of the F7 gene and genotype-phenotype relationship in patients with FVII deficiency. DNA sequencing showed the identification of pathogenic variants in 68.5% of the cohort. Interestingly, five of which had never been reported before in the international databases. Thus, this is very important news to share to the scientific community and perhaps, help those who have not yet submitted their data into the public database.
Russia is such a big country and I am wondering if they could narrow down the 54 cases to a specific region(s) in Russia. Did the majority of cases come from Moscow?
The authors have analysised their data set to a high and appropriate standard. It is interesting that they did not find any pathogenic variants in 31% of their cohort but they have presented a correct and logical argument as to why.
Abstract
Line 1
Missing a word. Sentence does not make sense:
Coagulation factor VII (proconvertin) is one of the proteins
Line 18
I do not understand this statement. Please clarify.
Four pathogenic 18 variants were early undescribed.
Line 55
Is the prevalence based on worldwide numbers?
Line 82
Add pH of TE Buffer and temperature frozen.
Table 1
Remove “Exon” from Column Heading and replace with “Target”
I am use to seeing primers listed with F = forward, R = Reverse. I”m not sure what the “D” is in reference too.
Line 98
Add ethidium bromide conc used and staining time.
Table 2
Make more concise. I note that there is a lot of repetition.
References
A range of relevant sources from 1982 to 2022 are cited.
Not all the doi’s are cited. e.g. Reference 19
https://doi.org/10.1182/blood.V92.5.1639
The grammar in this article needs to be improved. Either the introduction of comma’s, or words are missing to make your sentences or meanings complete.
Author Response
Thank you very much for taking the time to review our manuscript. Please find the detailed responses below and the corresponding corrections in track changes in the resubmitted files.
Comment 1: Russia is such a big country and I am wondering if they could narrow down the 54 cases to a specific region(s) in Russia. Did the majority of cases come from Moscow?
Response 1: Thank you for your question. We added this information in Materials and methods: "As in our country we have issues with diagnostics of rare diseases in regions, the majority of studied patients were from Moscow and Moscow region (N=35, 65%), other 14 patients (26%) were from European part of Russia and only 5 patients (9%) represented other regions." And we also added a few sentences in the discussion and conclusion about this bias.
Comment 2: Abstract Line 1 Missing a word. Sentence does not make sense: Coagulation factor VII (proconvertin) is one of the proteins
Response 2: Thank you, we corrected it.
Comment 3: Line 18 I do not understand this statement. Please clarify. Four pathogenic 18 variants were early undescribed.
Response 3: Thank you, we write this sentence and corrected the mistake with the number of novel mutations.
Comment 4: Line 55 Is the prevalence based on worldwide numbers?
Response 4: Yes, it based on worldwide numbers, we added it.
Comment 5: Line 82 Add pH of TE Buffer and temperature frozen.
Response 5: Thank you for your comment, we added this information to the Material and methods.
Comment 6: Table 1 Remove “Exon” from Column Heading and replace with “Target”. I am use to seeing primers listed with F = forward, R = Reverse. I”m not sure what the “D” is in reference too.
Response 6: Thank you, we corrected the column name. We use D as direct primer, but we changed their names to F.
Comment 7: Line 98 Add ethidium bromide conc used and staining time.
Response 7: Thank you, we added this information to the Materials and methods.
Comment 8: Table 2 Make more concise. I note that there is a lot of repetition.
Response 8: Sorry, but we do not understand what you have meant. This table describes pathogenic variants found in the study, repetitions can't be excluded, because mutations were found in the same exons and domains. If you clarify your point and suggest how to improve it, we gladly do it.
Comment 9: References A range of relevant sources from 1982 to 2022 are cited. Not all the doi’s are cited. e.g. Reference 19 https://doi.org/10.1182/blood.V92.5.1639
Response 9: Thank you very much for your comment, we looked for more doi again and added them to references. Now all articles have it.
Comment 10: Comments on the Quality of English Language. The grammar in this article needs to be improved. Either the introduction of comma’s, or words are missing to make your sentences or meanings complete.
Response 10: Thank you very much for your comment, we checked the manuscript one more time and made changes throughout the text.